# Targeting Mitochondria with ClpP Agonists as a Novel Therapeutic Opportunity in Breast Cancer

**DOI:** 10.3390/cancers15071936

**Published:** 2023-03-23

**Authors:** Rohan Wedam, Yoshimi Endo Greer, David J. Wisniewski, Sarah Weltz, Manjari Kundu, Donna Voeller, Stanley Lipkowitz

**Affiliations:** Women’s Malignancies Branch, Center for Cancer Research, National Cancer Institute, National Institutes of Health, Bethesda, MD 20892, USA

**Keywords:** breast cancer, mitochondria, cancer metabolism, ClpP

## Abstract

**Simple Summary:**

Mitochondria, often called “the powerhouse of the cell”, generate most of the energy required for all cellular processes. Mitochondria also provide a platform for multiple functions, including building essential macromolecules (nucleotides, proteins, and lipids), cell signaling, detoxification, and cell fate control. Studies have demonstrated that mitochondrial function is widely reprogrammed in breast cancers to promote tumor progression. Therefore, targeting mitochondria is currently one of the focal points in breast cancer research. Over 1000 proteins are involved with mitochondrial functions. The mitochondrial protease called ClpP plays a central role in mitochondrial protein quality control. Recently, ClpP agonists have emerged as a novel class of mitochondria-targeting drugs. Hyperactivating ClpP induces uncontrolled, but selective, degradation of ClpP substrates and disrupts mitochondrial functions, leading to growth inhibition of breast cancer cells, without adverse effect in non-malignant cells. The unique characteristics and mechanism of action of ClpP agonists provide new opportunities in breast cancer treatment.

**Abstract:**

Breast cancer is the most frequently diagnosed malignancy worldwide and the leading cause of cancer mortality in women. Despite the recent development of new therapeutics including targeted therapies and immunotherapy, triple-negative breast cancer remains an aggressive form of breast cancer, and thus improved treatments are needed. In recent decades, it has become increasingly clear that breast cancers harbor metabolic plasticity that is controlled by mitochondria. A myriad of studies provide evidence that mitochondria are essential to breast cancer progression. Mitochondria in breast cancers are widely reprogrammed to enhance energy production and biosynthesis of macromolecules required for tumor growth. In this review, we will discuss the current understanding of mitochondrial roles in breast cancers and elucidate why mitochondria are a rational therapeutic target. We will then outline the status of the use of mitochondria-targeting drugs in breast cancers, and highlight ClpP agonists as emerging mitochondria-targeting drugs with a unique mechanism of action. We also illustrate possible drug combination strategies and challenges in the future breast cancer clinic.

## 1. Introduction

Breast cancer is the most diagnosed cancer worldwide (excluding skin cancer), accounting for 12% of all new cancer cases annually [1]. For women in the US, breast cancer is the most commonly diagnosed malignancy (excluding skin cancer) (31%), and the second leading cause of cancer mortality overall after lung cancer [2]. Breast cancer is the leading cause of cancer death among African American and Hispanic women [3,4]. Incidence rates of breast cancer continue to rise by ~0.5% per year, although the cancer mortality rates have declined steadily since their peak in 1989, albeit at a slower pace in recent years (1.3% annually from 2011 to 2020) [3]. Early diagnosed breast cancers are often effectively treated using surgery, radiation, chemotherapy, endocrine therapy, and targeted therapies [5]. Breast cancers can be classified into distinct subtypes: estrogen receptor positive (ER+), human epidermal growth factor receptor 2 (HER2) amplified, and triple-negative breast cancer (TNBC) defined by the absence of ER and progesterone receptor (PR) expression and HER2 amplification [6]. TNBC lacks specific molecular markers and is usually more aggressive compared with other subtypes. Endocrine therapies for ER+, and HER2-targeted drug therapies for HER2 amplified subtypes have led to significant improvement of the survival rate for both early stage and advanced cancers [7]. TNBC therapy remains challenging with high rates of relapse and metastatic disease, despite significant improvements in chemotherapy and targeted therapies (e.g., trop2-targeted therapy and immune checkpoint inhibitors [ICIs]). There is an urgent need to explore more effective and targeted strategies to overcome such drug resistance and metastasis in current therapy, especially in TNBC [8,9,10,11,12].

Much effort has been made to uncover underlying mechanisms that drive the proliferation and metastasis of cancers. As a result, it has now become clear that cancer is not only a genetic disease but also a metabolic disease, where oncogenic signaling pathways dysregulate (or co-opt) energy regulation and anabolism to support rapidly spreading tumors [13,14,15,16]. Metabolic reprogramming has recently emerged as one of the major hallmarks of cancer [17,18,19]. Metabolism is rewired in cancer cells to meet the high demand of energy to promote proliferation. For instance, uptake of glucose and amino acids is increased to fulfill the biosynthetic demands; the demand for carbon and nitrogen is upregulated for biosynthesis of macromolecules such as nucleotides, proteins, and lipids; nicotinamide adenine dinucleotide phosphate (NADPH) production is elevated to support reductive metabolic reactions and to maintain redox balance [20]. These complex metabolic changes are intricately orchestrated by the mitochondria, a master regulator of cell metabolism.

In this review, we will first discuss the recent understanding of the role of mitochondria in breast cancers and analyze a wealth of evidence supporting the rationale that mitochondria are an emerging therapeutic target in breast cancers. We then examine the status of the development of mitochondria-targeting drugs, with special highlight of the caseinolytic mitochondrial matrix peptidase proteolytic subunit (ClpP) as an emerging drug target in breast cancers.

## 2. Mitochondria and Breast Cancer—An Overview

The mitochondrion is a double-membrane-bound organelle found in the cytoplasm of a majority of eukaryotic cells [21]. Mitochondria contain their own genome called mitochondrial DNA (mtDNA), a circular DNA of 16,569 bp in human, which encodes 13 subunits of the electron transport chain (ETC) complexes, 2 ribosomal RNAs (12S and 16S), and 22 transfer RNAs. Mitochondria are often called the ‘powerhouse’ of the cell, as they generate chemical energy in the form of adenosine triphosphate (ATP) via oxidative phosphorylation (OxPhos) at the ETC. Mitochondria also play pleiotropic roles as a center of macromolecule synthesis, such as nucleic acids, amino acids, and lipids. Mitochondria also regulate apoptosis, redox homeostasis, calcium homeostasis, innate immunity, cell signaling, and epigenetic modification of the nuclear genome [22,23,24,25,26,27,28,29,30].

Outside of OxPhos in mitochondria, ATP is generated via glycolysis, in which glucose is used and metabolized to pyruvate and lactate. From an energy efficiency perspective, OxPhos is a more efficient way to produce ATP (36 mol ATP/mol glucose) compared with glycolysis (2 mol ATP/mol glucose). Normal mammalian cells generate ATP from either OxPhos or glycolysis, in the presence or absence of O_2_, respectively. On the other hand, it has long been viewed that cancer cells primarily rely on glycolysis even in the presence of O_2_, due to “defective” mitochondria, called the “Warburg effect” [31,32]. In accordance with this view, some studies indicated that downregulation of OxPhos genes is associated with poor clinical outcomes in multiple cancer types [33], and that impaired OxPhos is linked to reduced or mutated mtDNA in cancers [34,35].

In recent years, however, our understanding of the “Warburg effect” has evolved to a more complex cognition of tumor metabolism. The recent development of quantitative omics tools such as RNAseq, proteomics, and metabolomics has provided compelling evidence to support that tumors, including breast cancers, are metabolically heterogeneous and flexible, depending on multiple factors such as cell lineage, differentiation state, and microenvironment [36,37]. In addition, cancer cells have a metabolic demand beyond ATP, such as nucleic acids, amino acids, and lipids to support growth [32,38]. A large body of evidence shows that mitochondria are not defective, but rather essential to cancer cells [39,40]. Herein, we discuss the recent findings with regard to how mitochondria contribute to breast cancer progression.

## 3. Evidence That Mitochondria Are a Target in Breast Cancer

### 3.1. OxPhos Is Upregulated in Breast Cancers

A study which analyzed ~2000 breast cancer patients showed that an OxPhos gene signature is significantly upregulated in human breast tumors (*p* < 1 × 10^−20^) relative to normal breast tissue, in both ER+ and ER- breast cancers [41]. An analysis from TCGA also showed that OxPhos genes are significantly increased in breast tumor samples compared to normal samples (*p* = 3.11 × 10^−7^) [42]. In ER-negative breast cancers, the OxPhos gene signature was associated with a ~30% increase in metastasis over a 10-year follow-up period (50% or 21.4% developed metastasis with or without the signature, respectively) [41]. Moreover, a recent study identified OxPhos as the top pathway upregulated in micrometastases of breast cancers, and pharmacological inhibition of OxPhos attenuated lung metastasis, highlighting the potential of OxPhos as a therapeutic target to prevent metastasis in breast cancers [43].

### 3.2. Breast Cancer Stem Cells (CSCs) and Circulating Tumor Cells (CTCs) Are Mitochondria-Dependent

CSCs and circulating tumor cells CTCs are responsible for tumorigenesis, chemoresistance, metastasis, and relapse [44,45]. In primary tumors, the majority of cancer cells are highly proliferative, differentiated, non-stem cells. In contrast, CSCs are a rare population (<1%) of cancer cells harboring an undifferentiated, slow-cycling phenotype, with self-renewal and tumor-initiating capacities [46,47,48,49]. These cells show distinct metabolic preferences; the bulk breast cancer cells rely on glycolysis, while quiescent breast CSCs depend on OxPhos, and proliferating CSCs utilize both OxPhos and glycolysis. CSCs also utilize metabolites generated from tumor stromal cells in what is called a “Reverse Warburg effect” [37,50,51,52,53] (Figure 1, left). OxPhos is shown to be critical for breast CSCs function [54,55,56,57,58,59,60,61,62,63]. For instance, inhibition of OxPhos by depleting mtDNA reduced an anchorage-independent survival and propagation of breast CSC-like cells [55,56,57,58]. MtDNA in exosomes mediates endocrine therapy resistance [59]. Samples from chemoresistant TNBC showed high dependency on OxPhos [60,61].

CTCs are shed from the primary tumor and intravasate into the blood circulation and mediate tumor metastasis [64]. CSCs and CTCs share similar phenotypes in breast cancers [65]. Peroxisome proliferator-activated receptor gamma coactivator 1-alpha (PGC1α), the master regulator of mitochondrial biogenesis, is overexpressed in breast CTCs [66]. OxPhos is elevated in breast CTCs, along with purine/pyrimidine metabolism, ROS, and antioxidant mechanisms [66,67,68]. Lung/bone-metastatic breast cancer cells show high PGC1α expression and mitochondrial metabolic activity [66,69,70]. In brain-metastatic breast cancers, OxPhos, the TCA cycle, and the pentose phosphate pathway (PPP), glutathione, and glycolysis pathways were all elevated [71] (Figure 1, right). A recent clinical study also provided the evidence that OxPhos contributes to breast cancer metastasis. The copper chelating agent tetrathiomolybdate (TM) inhibited lung metastasis and improved survival benefit in TNBC patients (NCT00195091) [72]. TM’s anti-tumor effect is shown to be an inhibition of OxPhos by copper depletion [73]. Thus, mitochondria are critical for breast CSCs/CTCs functions [50,52,74].

### 3.3. Mitochondrial Metabolism in Breast Cancers Is Widely Reprogrammed to Promote Their Growth

In addition to high demands of ATP, breast cancers heavily rely on multiple mitochondrial metabolic pathways. Many mitochondrial enzymes are overexpressed as discussed below and indicated shown in Appendix A.

ETC: The de novo pyrimidine synthesis is coupled to the respiratory chain and mediated by dihydroorotate dehydrogenase (DHODH) [75]. DHODH is overexpressed in human breast cancer tissues [76]. Ubiquinol-cytochrome C reductase core protein II (UQCRC2), a subunit of complex III, is also elevated in breast cancers [77].

Glutamine metabolism: Glutamine serves as a nitrogen donor for protein and nucleotide synthesis and maintains redox homeostasis and detoxication [78]. Glutaminase (GLS) is an enzyme that converts glutamine to glutamate. Glutamate is further converted to alpha-ketoglutaric acid (α-KG) via multiple enzymes, including glutamate dehydrogenase 1 (GLUD/GDH1). GLUD is upregulated in breast cancers [79]. α-KG is a key metabolite that drives the TCA cycle. The TCA cycle produces reducing power of nicotinamide adenine dinucleotide (NADH), which is utilized at the ETC to generate ATP. The TCA cycle also contributes to the biosynthesis of macromolecules, such as amino-acids, nucleotides, and lipids [80,81]. GLS1/GLS2 is pro-tumorigenic [82]. Glutamine metabolism is promoted by PGC1α in HER2+ breast cancer [83]. TNBC are known to be “glutamine-addicted”, with higher expression of key enzymes mediators of glutamine metabolism, including GLS, amino acid transporter-2 (ASCT2) [84,85,86,87,88,89]. Glutamate is enriched in breast cancer tissue [90]. The expression of glutamine synthetase (GS) and asparagine synthetase (ASNS) is correlated with poor prognosis in breast cancers [91,92].

The TCA cycle: High expression of dihydrolipoamide S-succinyltransferase (DLST/2-oxoglutarate dehydrogenase complex [OGDC]) and malic enzyme (ME2) is associated with poor prognosis in breast cancers [93,94].

Proline metabolism: Proline is derived from glutamate, and contributes to synthesis of arginine, glutamate polyamines through the urea cycle. Proline and its precursor delta-1-pyrroline-5-carboxylate (P5C) shuttle between the mitochondria and the cytoplasm, forming the “proline cycle”, thereby contributing to nucleotide synthesis, collagen synthesis, redox homeostasis by coupling with PPP. Proline biosynthesis is elevated in breast cancers [95,96], and pyrroline-5-carboxylate reductase (PYCR) 1, which mediates proline synthesis from P5C, is elevated in invasive breast cancers [97,98]. Proteomic analysis showed that PYCR1, PYCR2 and P5C synthase (P5CS/ALDH18A1) are all elevated in breast cancer patients’ samples [99]. Proline dehydrogenase (PRODH), which converts proline to P5C, is overexpressed in breast cancer metastases compared with primary tumors [100].

Serine, glycine and folate-mediated one carbon cycle: Serine and glycine are essential metabolites for tumor growth, and the related enzymes are upregulated in breast cancers [101,102,103,104]. Phosphoglycerate dehydrogenase (PHGDH), the first rate-limiting enzyme in the serine synthesis pathway, is elevated in ~70% of ER- breast cancers, and PHGDH inhibition showed an anti-proliferative effect [103]. PHGDH expression was elevated in 90% of brain metastases of breast cancer [105]. Serine serves as a major donor of one-carbon units to the folate cycle, which is required for de novo purine/pyrimidine synthesis and production of NADPH [106,107]. Serine hydroxymethyltransferase 2 (SHMT2), an enzyme that converts serine to glycine, is shown as a prognostic marker and therapeutic target for breast cancers [108]. In the folate cycle, methylenetetrahydrofolate dehydrogenase (MTHFD) produces purines. High MTHFD2 is correlated with worse prognosis in breast cancers [109,110]. Thymidylate synthase (TYMS/TS), the target of 5′-fluorouracil, is a rate-limiting enzyme in thymidylate biosynthesis [111]. TYMS expression was elevated in breast cancer compared with normal tissues, and was greater in TNBC than non-TNBC, and higher TYMS was associated with poor prognosis [112]. Consistent with this, purine/pyrimidine metabolism enzyme gene sets are elevated in TNBC [89]. Dihydrofolate reductase (DHFR), another enzyme in folate metabolism, is essential to purine synthesis and contribute to mitochondrial thymidylate biosynthesis. DHFR is a target of methotrexate (MTX). The DHFR activity is elevated in MTX-resistant breast cancer cells [113].

Fatty acid synthesis (FAS): Elevated FAS is one of the most common metabolic traits in cancers [114,115]. FAS occurs in the cytoplasm via lipogenic enzymes, such as acetyl-CoA carboxylase 1 (ACC1/ACACA) and fatty acid synthase (FASN), using acetyl-CoA as the substrate. ACC1 is involved with breast cancer metastasis [116]. FASN is overexpressed in 70% of operable TNBC and is associated with poor prognosis [117,118]. FASN regulates Bcl-2 family proteins and controls breast cancer cell survival [119]. Acetyl-CoA synthetase 2 (ACSS2) is amplified in breast cancers and targeting ACSS2 impaired tumor growth in TNBC [120,121]. ATP citrate lyase (ACLY), an enzyme involved with de novo synthesis of cholesterol and FA, is overexpressed in breast cancer [122].

Fatty acid oxidation (FAO): FAO is the mitochondrial aerobic process of breaking down fatty acid (FA) into acetyl-CoA-units [123]. Overexpression of carnitine palmitoyl transferase I (CPTI), a rate-limiting enzyme of FAO, is associated with breast cancer progression [124,125,126]. FAO contributes to elevated ATP in TNBC and metastasis [127,128], and promotes apoptosis-resistance of TNBC by increasing mitochondrial membrane lipids [129]. FAO is also critical for breast CSCs function [130], and CD36, a FA receptor, is linked to breast cancer metastasis [131].

The mevalonate pathway: The mevalonate pathway synthesizes cholesterol from acetyl-CoA. High cholesterol is associated with breast cancer metastasis, and cholesterol biosynthesis is elevated in breast CSCs [132,133,134,135]. Statin use is associated with lower mortality in breast cancers [136]. The mevalonate precursor enzyme 3-hydroxy-3-methylglutaryl-CoA synthase 1 (HMGCS1) is overexpressed in breast cancer, and a key mediator of enrichment of breast CSCs [137,138]. High expression of HMGCS1 and 3-hydroxy-3-methylglutaryl-coenzyme A reductase (HMGCR) is correlated with poor prognosis in breast cancer [139]. Together, these data suggest that many metabolic pathways within or linked to the mitochondria are widely dysregulated in breast cancers, and those alterations collectively promote tumor growth and metastasis.

### 3.4. Myc, TP53, PIK3CA, Bcl-2 Family Proteins Control Mitochondrial Metabolism in Breast Cancers

The Myc proto-oncogene regulates multiple cellular processes. It is well established that Myc promotes breast tumorigenesis and progression [140,141,142,143]. Myc is amplified in breast cancers (~25% of all breast cancers, ~50% in TNBC) and is associated with poor prognosis [144,145,146,147]. Myc is a major driver of mitochondrial biogenesis [148,149]. Among ~1000 nuclear-encoded human mitochondrial genes, over 500 genes are Myc targets [150,151]. Myc also indirectly regulates mitochondrial gene expression via repression of microRNAs [152,153]. Myc globally participates mitochondrial metabolism [154]. Myc promotes glutamine metabolism [152,155,156], proline biosynthesis [96,157], serine biosynthesis [158] and FAO [159,160].

The tumor-suppressor gene TP53 regulates gene expression involved in cell cycle arrest, apoptosis, and senescence [161]. TP53 mutations (TP53mut) are observed in 25–30% of all breast cancers, ~80% in TNBC and 72% in HER2 amplified subtypes [162]. TP53mut participate metabolic changes via numerous mechanisms [163,164]. For instance, TP53mut enhances OxPhos [165,166] and FAS [167]. It also elevates the transcriptional activity of sterol regulatory element-binding proteins (SREBP) family, supporting the mevalonate pathway and cell growth [168].

Phosphatidylinositol-4,5-bisphosphate 3-kinase catalytic subunit alpha (PIK3CA) mutation (PIK3CAmut) leads to increased phosphatidylinositol 3-kinase (PI3K) activity. PIK3CAmut is observed in ~40% of ER+/HER2-breast cancer patients [169]. PIK3CAmut potentiates de novo lipid synthesis, pyruvate entry into the TCA cycle, and glutamine metabolism via the PI3K-AKT serine/threonine kinase (AKT) signaling pathway in breast epithelial cells [170,171,172].

The Bcl-2 family proteins are apoptosis regulatory proteins functioning at the mitochondria [173]. The anti-apoptotic Bcl-2 proteins (Bcl-2, Bcl-xL, Bcl-w, and Mcl-1) restrain pro-apoptotic proteins (Bax, Bak), thus preserving mitochondrial outer membrane integrity, preventing release of pro-apoptotic proteins, thereby inhibiting apoptosis. Bcl-2 is overexpressed in breast cancers (~75%) [174]. Bcl-xL is also overexpressed (43%) and associated with metastasis and higher tumor grade [175,176]. Overexpression of Mcl-1 is even more frequent and is associated with poor prognosis, and TNBC are dependent on Mcl-1 [177]. The Bcl-2 family proteins mediate therapy resistance in breast cancers [178]. Bcl-2 and Bcl-xL also regulate mitochondrial dynamics [179] and support metabolic robustness by elevating NAD(P)H and ATP in breast cancer cells [180,181].

### 3.5. Reactive Oxygen Species (ROS) Plays a Dual, Complex Role in Breast Cancer

Mitochondria are the major source of ROS, and also provide ROS-scavenging systems, such as glutathione, superoxide dismutase, and NADPH [182,183]. Thus, mitochondria play a major role in redox homeostasis in a cell [25,184]. ROS has a dual, complex role in cancers; either pro- or anti-tumorigenic, depending on the context [185,186]. High ROS is often observed in aggressive, drug-resistant cancers, but it can be lethal; therefore, cancer cells rewire the ROS-scavenging systems [115,187,188,189,190,191,192]. TNBC exhibit higher ROS and higher antioxidant capacity than non-TNBC and are dependent on ROS for cell survival [193,194]. High ROS causes mitochondrial damage and further generates ROS as a “vicious cycle” [195,196]. High ROS stimulates the PI3K-AKT-mammalian target of the rapamycin (mTOR) pathway, leading to the activation of Myc, Ras-mitogen-activated protein kinase (MAPK), the NF-kB pathway, and anti-apoptotic Bcl-2 family proteins, promoting cell survival and metastasis [188]. ROS also triggers DNA mutations, leading to cellular senescence, increased inflammatory cytokine release, and activation of the innate immune system [29,197,198,199,200]. In addition, low levels of ROS are critical for CSCs to maintain their self-renewal ability [25,187,201,202,203]. Breast CSCs have a lower level of ROS compared with bulk tumors, and pharmacologic inhibition of ROS scavengers in breast CSCs decreased their clonogenicity [204]. Similarly, exposure to H_2_O_2_ treatment impaired tumor formation ability in breast CSCs and led to senescence [205]. Thus, either a high or low level of ROS can enhance breast cancer growth via different mechanisms.

### 3.6. Additional Mechanisms by Which Mitochondria Contribute to Breast Cancer Progression

The tumor microenvironment (TME) consists of tumor cells and their neighbor cells, such as cancer-associated fibroblasts (CAFs), immune cells, and the extra cellular matrix. Interactions between these components of the TME promote tumor progression [206]. While understanding the complex nature of breast TME is challenging, a few studies indicate the role of mitochondria in breast TME. Breast cancer cells hijack the mitochondria from immune cells via nanotubes, and inhibition of the nanotube assembly and an ICIs showed the anti-tumor effect [207]. PYCR1 is elevated in the stroma and CAFs in breast cancers, and inhibition of PYCR1 reduced collagen synthesis, tumor growth, and metastasis [208]. Free fatty acids produced by tumor-surrounding adipocytes are used by breast cancer cells to promote FAO and cancer invasion [209].

Dysregulated epigenetic processes contribute to therapy resistance in breast cancers [210]. Mitochondrial metabolites drive epigenetic changes, such as histone acetylation, methylation, and DNA methylation [211]. A study showed that oncometabolite 2-hydroxyglutarate (2-HG) is elevated in a subset of breast cancers, and high 2-HG and a distinct DNA methylation pattern were associated with poor prognosis [212]. Mechanistically, Myc stimulates GLS1 activity, and the increased glutamine consumption promotes aberrant 2-HG accumulation, leading to DNA methylation in breast tumors [212].

The ATP-driven efflux transporters contribute to chemoresistance [213]. ATP-binding cassette (ABC) transporters utilize mitochondria derived-ATP as a source of energy to efflux drugs out of cancer cells, leading to chemo-resistance [214].

Mitochondrial undergo continuous cycles of fission and fusion, and these dynamics are closely related to its function [215,216,217]. Studies indicate a role of mitochondrial dynamics in breast cancer, although there are contradictory findings. Mitochondrial fission is associated with either poor outcomes [218,219] or anti-tumor activity [220,221].

In summary, mitochondria and breast cancer cells maintain a symbiotic relationship via multiple mechanisms that are intricately interconnected to each other to further promote breast cancer survival and growth (Figure 2). Additional mitochondria-associated factors not discussed in this review, such as mitophagy and ROS-induced senescence [222,223], may also support breast cancer progression. Thus, targeting mitochondria is reasonable in breast cancers, and numerous drugs have been developed [224,225]. Next, we will discuss the status of ClpP agonists, a novel class of mitochondrial drugs.

## 4. ClpP

Because of the interest in targeting mitochondria, a number of drugs targeting different aspects of mitochondria have been developed. In the rest of the review, we will focus on ClpP agonists. A summary of other drugs targeting other mitochondrial pathways are discussed in the Appendix A.

### 4.1. ClpP—The Structure and Functions

In humans, mitochondria contain over ~1000 proteins that fulfil multiple functions [226,227]. According to the latest version of MitoCarta 3.0, an inventory of mammalian mitochondrial proteins, a total of 1136 human genes, encode mitochondrial proteins [150]. Among them, 99% proteins are encoded by nuclear DNA, and 13 proteins are encoded by mtDNA. The sub-mitochondrial localizations of these proteins are the mitochondria matrix (46%), the mitochondrial inner membrane (IMM, 32%), the mitochondrial outer membrane (OMM, 10%), the mitochondrial intermembrane space (IMS, 5%), the mitochondrial membrane (3%), and unknown (5%). These proteins participate in multiple mitochondrial pathways including metabolism (40.6%), mtDNA maintenance, mtRNA metabolism and translation (20%), OxPhos (14.9%), protein import, sorting and homeostasis (7.6%), mitochondrial dynamics (9.1%), small-molecule transport (7.5%), and signaling (4.2%) [150].

Mitochondrial proteins undergo processing events, form functional assemblies, and localize to correct locations. During this process, unfolded and misfolded proteins are cleared to maintain mitochondrial protein homeostasis, which is critical for mitochondrial function [228]. Misregulation of mitochondrial proteostasis leads to human diseases, including cancers [229,230,231,232], and specialized molecular chaperones and proteases conduct this maintenance [233]. At least 45 proteases are present in the different compartments of a mitochondrion [234]. Among these proteases, the evolutionally conserved ATP-dependent proteases (AAA+ [ATPases Associated with various cellular Activities] superfamily) represent core components of the mitochondrial proteolytic system performing mitochondrial protein quality control in a mammalian cell. The members of this AAA+ proteases family are the Lon protease 1 and the ClpXP complex in the mitochondria matrix, and the i-AAA and m-AAA proteases in the inner membrane [234,235,236] (Figure 3). Among the four AAA+ proteases, ClpXP is most extensively studied in terms of its biochemical and genetic features, crystal structures, and the molecular basis of protein substrate recognition [237].

ClpXP consists of the caseinolytic mitochondrial matrix peptidase chaperone subunit X (ClpX; AAA+ ATPase) and ClpP (a tetradecameric peptidase). ClpX is a hexametric ATP-dependent protein unfoldase and translocase [238]. ClpP is a barrel-like peptidase assembled from two stacked heptameric rings, which enclose a roughly spherical proteolytic chamber [239] (Figure 4A). The proteolytic activity of ClpP is usually tightly regulated by ClpX [240]. ClpX recognizes degrons on proteins to be degraded and provides energy to unfold and translocate linearized protein into the barrel of ClpP [241]. Substrate specificity is not defined by the amino acid sequence, rather ClpX recognizes specific degradation signals and interacts with certain adaptor proteins [242]. ClpX then feeds the unfolded substrate through the axial pore to the proteolytic chamber of ClpP, and the substrate is cleaved into small peptide fragments [242,243] (Figure 4B). ClpX binds to one or both ends of ClpP and this allows for both single- and double-capped ClpXP complexes [237]. As with bacterial ClpP, human ClpP cylinders switches dynamically between extended and compact forms (Figure 4C). The active, extended form is required for substrate degradation, while an inactive compact state allows peptide product release from the ClpP inner chamber [243,244,245,246].

The role of ClpP as a master regulator of mitochondrial protein quality control and homeostasis is well established. Many mitochondrial proteins are identified as substrates of ClpXP, including proteins involved in ETC, the TCA cycle, mitochondrial ribosome, mitochondrial gene transcription and translation, glutamine metabolism, and folate metabolism [58,247,248,249]. By degrading misfolded or damaged proteins, ClpXP maintains the integrity of mitochondrial functions.

ClpP also serves as a central mediator of the mitochondrial unfolded protein response (UPR^mt^) [250,251], a conserved transcriptional response activated by multiple forms of mitochondrial dysfunction and regulated by mitochondrial-nuclear communication. The UPR^mt^ supports the recovery of mitochondria from damage, restores ETC function, eliminates excess amounts of ROS, and promotes cell survival [252,253]. Recent studies indicated the prolonged UPR^mt^ contributes to organismal deterioration including cancer [196,253,254]. In breast cancers, elevated expression of UPR^mt^-related genes is significantly associated with poor overall and metastasis-free survival [255]. Activated UPR^mt^ was observed in HER2 amplified breast cancer tissues [256].

ClpP is predominantly expressed in non-malignant tissues with high mitochondrial content such as skeletal muscle, liver, and heart, suggesting its critical role in normal tissue [257]. Despite the expression in such critical organs, mice with homozygous deletion of the CLPP gene are viable, but infertile, and exhibit hearing loss and growth retardation with accumulation of ClpX and mtDNA [258]. Similarly, recessive CLPP mutations in the human CLPP genes are linked to a rare genetic disease Perrault syndrome, which shows infertility and hearing loss [259,260].

### 4.2. ClpP Activation Impairs Breast Cancer Cell Viability

While the exact roles of ClpP in tumorigenesis are not clearly established, emerging evidence demonstrate that ClpP is overexpressed in multiple malignancies including breast cancers [261,262,263]. Recent analysis of TCGA database also confirmed that ClpP is overexpressed in breast and other cancers [264]. In addition, a high level of ClpP expression is associated with the disease stage, metastasis, poor prognosis and is proposed as a prognostic marker in breast cancers [261,265,266]. Thus, ClpP is considered an emerging target for breast cancer.

Interestingly, both inhibition and hyperactivation of ClpP can lead to impaired OxPhos, resulting in cancer cell death [267]. Inhibition of ClpP induces the accumulation of damaged and misfolded mitochondrial proteins, whereas hyperactivation of ClpP leads to unregulated degradation of ClpP substrates, and both mechanisms can elicit defective cellular respiration and eventual cell death [233]. Importantly, whether inhibition or activation of ClpP induces cell death appears to be context dependent. CLPP is an essential gene for cell survival in leukemia cells and ClpP inhibition is lethal in acute myelogenous leukemia (AML), chronic myelogenous leukemia, and osteosarcoma [262]. In contrast, ClpP activators show anti-tumor effect in breast, ovary, colorectal, glioblastoma, and other cancers [58,268,269,270,271,272]. In breast cancer cells, transient knockdown of CLPP by siRNA induced apoptosis and inhibited cell viability, migration, invasion in breast cancer cells [265], while CRISPR/Cas9-mediated deletion of CLPP gene did not affect cell viability presumably due to cellular adaptation [58,270]. In contrast, activation of ClpP by ClpP agonists exerts an anti-tumor effect in breast cancer cells [58,270]. This suggests that ClpP activation, but not ClpP inhibition, is the better targeting approach in breast cancers.

### 4.3. ClpP Activating Drugs

Since ClpP activators, but not ClpP inhibitors, show cytotoxicity in breast cancers, we will focus on ClpP activators in this review (for ClpP inhibitors, we refer readers to other excellent reviews [233,264,266]. There are three classes of ClpP agonists: acyldepsipeptide antibiotics (ADEPs), ONC201 and its analogs (imipridones), and TR compounds. The representative ClpP agonists and structures are illustrated in Figure 5.

ADEPs: ClpP has emerged as an anti-bacterial target during the mechanism of action studies on acyldepsipeptide antibiotics (ADEPs) in last two decades. ADEPs were first reported as antibiotics [273]. Identification of the resistance-mediating mutation within an ADEP-resistant Escherichia coli mutant and affinity chromatography with an immobilized ADEP analog led to identification of ClpP as the direct target of ADEPs [274]. ADEPs increase the activity of the bacterial ClpP, leading to bacterial cell death. As ClpP plays important roles in the bacterial virulence due to the broad range of substrates, extensive effort has been made to develop ADEPs as novel therapeutics for drug-resistant bacteria [242,274,275,276,277]. Medicinal chemistry studies revealed the structure–activity relationship and yielded many derivatives with enhanced in vitro potency and stability [278,279,280,281]. The anti-bacterial efficacy of ADEPs has been demonstrated in lethal bacterial infections in rodent models [274,278,282]. Wong et al. were the first to demonstrate that ADEPs binds to human ClpP (HsClpP) using crystallographic structural analysis [283]. ADEPs induce caspase-dependent apoptosis, mitochondrial fragmentation, and OxPhos inhibition in HEK293 cells. At present, ADEP analogues are being tested in preclinical models in breast cancers, but the findings are limited.

ONC201 and its analogs (Imipridones): ONC201 (a.k.a., TIC10) is the first-in-class small molecule of the imipridone family [284]. ONC201 was identified in a chemical library screen to find a small molecule that transcriptionally induces tumor necrosis factor-alpha related apoptosis-inducing ligand (TRAIL), leading to an autocrine induction of apoptosis in the cancer cell. The proposed mechanism of action of ONC201 was that it inhibits extracellular signal-regulated kinase (ERK) and AKT, leading to the translocation of Foxo3a into the nucleus, where it activates transcription of the TRAIL gene [284]. Subsequent studies evaluated ONC201 as an antagonist of dopamine receptors (DRD)2/3 [285,286,287]. There were some preclinical data supportive of DRD2 antagonistic activity [286], and the clinical activity seen in diffuse intrinsic pontine gliomas (DIPG) has been suggested to be mediated through DRD2 inhibition [288,289]. However, this remains controversial. No direct evidence of ONC201 binding to DRD2/3 has been shown, gene suppression or deletion of DRD2/3 did not abrogate the ONC201’s cytotoxic effect [286], and DRD2 transcript is not detectable in many breast cancer cell lines that are sensitive to ONC201 [269]. This requires further evaluation to determine if any activity of ONC201 or other ClpP agonists is due to DRD2 antagonism.

While ONC201 was shown to increase Poly (ADP-ribose) polymerase 1 (PARP) cleavage and apoptosis in some cancer models [284,290], apoptosis was not observed in breast cancers [269,270]. Our group was the first to report that the cytotoxicity of ONC201 is due to targeting mitochondria and is independent of TRAIL-mediated apoptosis [269]. We found that ONC201 inhibits OxPhos, depletes ATP, mtDNA, multiple mitochondrial proteins, such as mitochondrial transcription factor A (TFAM), ETC proteins, accompanied with mitochondrial structural damage and an integrated stress response (ISR) indicated by ATF4 and CHOP induction [269]. We also found that breast cancer cells that lack functional mitochondria (e.g., rho0 cells) were ONC201-resistant. Our finding was confirmed and extended by two studies which demonstrated that ONC201 binds to and activates ClpP using crystallography structural analysis, mass-spectrometry, and affinity chromatography/drug competition assays [268,271]. Another group identified ClpP as the target of ONC201 using a genome wide CRISPR KO library screen [291]. Later, ONC206 and ONC212 were developed by Chimerix, Inc. (https://www.chimerix.com/, accessed on 2 January 2023) as more potent imipridones compared with ONC201. Studies confirmed that these imipridones impair OxPhos, induce mitochondrial damage, ROS, and ISR [271,292]. ONC201 has been shown to penetrate the brain-blood barrier [284].

TR compounds: TR compounds are a novel series of imipridone-derived compounds and related chemicals being developed by Madera Therapeutics, LLC (https://maderat.com/, accessed on 2 January 2023). Multiple TR compounds, including TR-57, TR-31 (=ONC212), were found to be ∼50–100-fold more potent compared with ONC201 in cytotoxicity in breast cancer cells. TR compounds induced an ISR at nanomolar concentrations, significantly lower compared with those induced by ONC201 at micromolar concentrations [268]. Affinity chromatography/drug competition assays demonstrated that the TR compounds bind to ClpP with ∼10-fold higher affinity compared to ONC201 [268]. Similar to ONC201, TR compounds reduce target protein levels (e.g., TFAM) and impair OxPhos [58,268,269].

The unique feature of all ClpP agonists described above is that they activate ClpP in the absence of ClpX (Figure 6 and Figure 7). The X-ray crystallography structure demonstrated that ADEP binds to the hydrophobic pockets of ClpP and dissociates the ClpXP complex at substoichiometric concentrations, leading to ClpP activation independent of regulatory subunit ClpX [283] (Figure 6A). Similar to ADEPs, imipridones directly binds to ClpP, and displace ClpX, and the bindings triggers the opening of channel-like pore of ClpP, and thereby increase its protease activity without the ClpX [233,271,293] (Figure 6B). Recently, X-ray crystallography demonstrated that TR compounds bind to ClpP, with enhanced binding affinities due to their greater shape and charge complementarity with the surface hydrophobic pockets of ClpP [294] (Figure 7C). Moreover, N-terminome profiling of MDA-MB-231 cell line upon treatment with one of TR compounds revealed the global proteomic changes and characterized the sequence and structural properties for protein cleavage generated upon TR compound-induced ClpP-dependent proteolysis of cellular proteins [294].

## 5. Preclinical Studies Using ClpP Agonists in Breast Cancers

Many preclinical studies have demonstrated the efficacy of ClpP agonists in breast cancer models (Table 1). Among ClpP agonists, ONC201 has been most extensively tested in various preclinical models, including breast cancers. TR compounds have also been tested recently in breast and other cancer cells. At present, data on ADEPs as a cancer therapeutic are limited. ADEP1 and ADEP2 showed anti-tumor activities in multiple renal cancer cells [295]. ADEP41 showed cytotoxicity in HeLa (cervical), U2OS (osteosarcoma), SH-SY5Y (neuroblastoma) at IC_50_ 0.5–0.9 uM, but the effect on breast cancers is not yet reported [283]. Several critical findings from the preclinical studies using ClpP agonists in breast cancer models are discussed below.

### 5.1. Findings from In Vitro Studies

#### 5.1.1. ClpP Agonists Uniquely and Broadly Impair Mitochondrial Function

ClpP agonists induce degradation of mitochondrial proteins (e.g., TFAM, mitochondrial elongation factor Tu [TUFM], and ETC complex proteins), depletion of mtDNA copy number, downregulation of OxPhos, loss of ATP, mitochondrial structural damage, and elevation of ROS level in multiple breast cancer cell lines [58,269,270,304]. Moreover, ClpP agonists dysregulate metabolic pathways not affected by other mitochondrial drugs such as metformin, oligomycin, CPI-613 [58]. ClpP agonists downregulate multiple enzymes; GLS, P5CS/ALDH18A1, PYCR1/2, NADK2/mitochondrial NAD+ kinase, ME2, SHMT2, MTHFD2, TYMS and HMGCS1. All of these enzymes are associated with metabolic rewiring that drives breast cancers as described above. The proline cycle, ME2, and folate cycle are involved with production of NAD(P)+/NAD(P)H; therefore, ClpP agonists also deplete NAD(P)+/NAD(P)H, leading to inhibition of macromolecule biosynthesis and ROS production [58].

#### 5.1.2. ClpP Agonists Inhibit Breast Cell Growth, Proliferation, Viability, but Do Not Induce Apoptosis

Studies testing ONC201 and/or TR compounds in breast cancer models collectively show that ClpP agonists exert a significant anti-proliferative effect accompanied with ISR and cell cycle arrest, rather than inducing apoptosis [58,268,270,298,299,301,302,304,306]. These observations are distinct from what has been reported as apoptosis observed in other malignancies, such as colon cancers and hematological malignancies [307,308,309]. Recent studies revealed that the anti-proliferative effect of ClpP agonists observed in breast cancers represent a senescence-like phenotype [304,310]. These findings suggests that the impact of ClpP targeting in cancers is context dependent.

#### 5.1.3. Non-Malignant Cells Are Resistant with ClpP Agonists

One notable finding from ClpP agonist studies is that ClpP agonists do not induce cytotoxicity in non-malignant cells. While ONC201 induces mitochondrial structural damage and depletes mtDNA in human foreskin fibroblast (HFF) cells, it does not impair cell viability [269]. Other studies also reported that ONC201 and its analogs do not exert cytotoxicity in normal human fibroblasts, such as MRC5, WI38, HFF1, 18Co, 19Lu [269,305], normal lung epithelial cells [311], and normal bone marrow cells [312]. The mechanisms of the resistance to ClpP agonists in normal cells remains elusive but intriguing, suggesting a beneficial feature of ClpP targeting therapies.

#### 5.1.4. The Subtype-Specific Effect of ClpP Agonists in Breast Cancer Remains Unclear

Several studies investigated ONC201 sensitivity with a specific focus on breast cancer subtypes. We tested ONC201 in 17 breast cancer cell lines consisting of different subtypes (4 ER+, 1 ER+/HER2+, 3 HER2+, 4 TNBC Basal A, and 5 TNBC Basal B), and the IC_50_ range was from 0.78 to 4.83 micromolar without any subtype-specific difference [269]. A study tested ONC201 in 10 TNBC cell lines and found that the IC_50_ range varied from 0.032 to 10.12 micro molar; however, mesenchymal-like TNBC (e.g., SUM159, SUM149, MDA-MB-157, MDA-MB-231, and BT549) and epithelial-like TNBC (e.g., HCC70, BT20, HCC1806, HCC1937, and MDA-MB-468) were equally sensitive to ONC201 [298]. Another study tested 17 TNBC cell lines and reported that the ONC201 IC_50_ ranged from 2.05 to 43.39 micromolar, and that HCC70 (Basal-like 1), MDA-MB-157 and SUM159 (Mesenchymal) were relatively resistant compared with MDA-MB-468 (Basal-like 1), CAL51 (Basal-like2) and that higher ClpP protein expression is correlated with a lower ONC201 IC_50_ in TNBC cell lines [302]. This proposed link between ClpP protein expression and ClpP agonist sensitivity needs to be further investigated, as the ClpP protein expression level does not appear to be correlated with sensitivity to ClpP agonists in 17 breast cancer cell lines with different subtypes (3 ER+, 1 ER+/HER2+, 3 HER2+, 5 TNBC Basal A, and 4 TNBC Basal B) in our study [58].

#### 5.1.5. Comparison of ClpP Agonist Potency in Representative Breast Cancer Cell Lines

The IC_50_ of ONC201 is in the micromolar range, while that of ONC206 is several folds lower, and ONC212 is more potent in the nanomolar range in breast cancer cells. Compared with imipridones, TR compounds show significantly higher potency; IC_50_ of TR-57 and TR-107 exert anti-proliferative effects in the 10–25 nanomolar in TNBC cells (Table 2). The superior potency of TR compounds over other ClpP agonists is attributed to the better binding properties to ClpP [294].

### 5.2. Findings from In Vivo Studies

#### 5.2.1. ClpP Agonists Showed an Anti-Tumor Effect in Animal Models of Breast Cancer

The anti-tumor activity of ClpP agonists in breast tumors has been demonstrated in multiple in vivo studies. In these studies, ONC201 was used at doses of 25–100 mg/kg, weekly, orally, or intraperitoneally (i.p.) (Table 2). ONC201 treatment (i.p. at 100 mg/kg, weekly) had an anti-tumor growth effect in MDA-MB-231 xenograft model [284]. Another study also showed that ONC201 ablated MDA-MB-231 xenograft tumors, and that dose intensification of ONC201 inhibited invasion, migration, and metastasis [300]. Anti-tumor growth effects of TR-107 in MDA-MB-231 xenograft models was also reported. TR-107 treatment (oral, 4 or 8 mg/kg, twice a day, twice a week) induced a ~50% reduction in tumor volume and extended the animal median survival ~35% compared to control. TR-107 treatment was well tolerated with less than 5% weight loss observed even at the highest dosing regimen. Importantly, the on-target in vivo activity of TR-107 was confirmed, as mtDNA copy number was reduced and ClpP substrate proteins such as TFAM, PYCR2, and HMGCS1 were all downregulated in tumor samples from the TR-107-treated group [270]. The inhibitory effect of ClpP on breast CSC function has also been shown in vivo. MDA-MB-231 cells pre-treated with either ONC201 or TR-57 failed to form tumor in mice, indicating that ClpP agonists impair CSC function [58]. Collectively, these studies demonstrated that ClpP agonists induce cytotoxicity in TNBC and inhibit breast CSC functions.

#### 5.2.2. Pharmacokinetics

The pharmacokinetics (PK) property of select ClpP agonists in mice from three independent studies is summarized in Table 3 [270,284,305]. In the initial study, the half-life of ONC201 was reported as 6.42 h (25 mg/kg, oral) [284], and that of ONC212 was 4.31 h (125 mg/kg, oral) [305], whereas a recent study showed the shorter half-life [270]; ONC201 was 0.31 h (10 mg/kg, oral), and that of ONC212 was 1.49 h (10 mg/kg, oral). The half-life of TR-57 was 1.4 h or 1.52 h (10 mg/kg, oral or 2 mg/kg, i.v., respectively), that of TR-107 was 0.9 h (10 mg/kg, oral). Despite similar half-life in mice models, the exposure (AUC) of TR-57 and TR-107 (10 mg/kg, oral) was 750–900-fold higher than that of ONC201 (10 mg/kg, oral). TR-57 showed high rapid absorption rate (F% of 61%) after oral administration. The protein-binding studies for TR-107 show a serum-free fraction of 10%. Collectively, both TR-57 and TR-107 show favorable PK characteristics compared with ONC201 and ONC212.

#### 5.2.3. ClpP Agonists Induce Natural Killer (NK) Cell Recruitment and Activation of Immune Cells

As discussed above, mitochondria participate in the modulation of TME in breast cancers. Studies have shown that ONC201 promotes NK cell recruitment to tumors and NK cells mediated killing of breast cancer cells [300,301]. It is also shown that ONC201 induces accumulation of CD4+/CD8+CD3+ T cells in syngeneic MC38 colorectal tumors [300]. Similarly, ONC206 treatment increased NK cells and activated CD4+ cells within ovarian tumors in a mouse model [313]. The impact of ClpP agonists on immune cells needs to be further investigated.

## 6. Clinical Trials of ClpP Agonists in Breast Cancers

ClpP agonists have been actively exploited in multiple clinical trials with various types of malignancies; however, clinical studies of ClpP agonists in breast cancers have not reported results. Among multiple ClpP agonists, ONC201 is the only ClpP agonist entered in clinical trials for breast cancers as of January 2023.

In the first-in-human phase 1/2 study of ONC201 in patients with advanced cancers (no breast cancer patients were included), ONC201 was orally administered once every 3 weeks defined as one cycle, with a dose escalation from 125 to 625 mg in 10 patients. ONC201 was well-tolerated, and the recommended phase 2 dose (RP2D) was determined as 625 mg, weekly. An additional 18 patients were treated at the RP2D in an expansion phase to collect additional safety, PK/PD information. PK analysis revealed a *C*_max_ of 1.5 to 7.5 μg/mL (~3.9–19.4 micromolar), the mean half-life ONC201 was 11.3 h and mean AUC was 37.7 h·μg/L [285,314]. No objective responses by RECIST were achieved; however, radiographic regression of several individual metastatic lesions was observed along with prolonged stable disease (>9 cycles) in prostate and endometrial cancer patients.

Table 4 summaries trials targeted to breast cancer patients. The phase 2 clinical trial of ONC201 in metastatic, previously treated ER+ and TNBC breast cancer patient was recently completed but has not yet published the results (NCT03394027). The dosing schedule in this study was 625 mg of the ONC201 taken orally, every 7 days in 28-day cycles. Unfortunately, ONC201 did not show clinical efficacy as reported at Clinicaltrials.gov (accessed on 20 January 2023).

ONC206, ONC212, and TR compounds are not yet tested in breast cancers. Another study intended to determine the objective response rate to ONC201 with a methionine-restricted diet in patients with metastatic TNBC, but was terminated due to the low number of patients enrolled (NCT03733119).

## 7. Challenges to Achieve Better Clinical Outcomes of ClpP Agonists Therapy

While ClpP agonist-induced mitochondria targeting therapy is an emerging and attractive approach, there are many questions to be addressed. Establishing predictive biomarkers that can identify patients who will be most effectively treated with ClpP agonists will be significantly beneficial to improving therapeutic outcomes. Pharmacodynamic (PD) biomarkers are also required to monitor the activity of ClpP agonists in patients. To further enhance the therapeutic efficacy, the combination of ClpP agonists with other drugs needs to be explored. In this section, we will discuss the research status in these challenges.

### 7.1. Predictive Biomarkers

Gene Expression: As discussed above, a higher ClpP expression level is correlated with a lower IC_50_ of ONC201 in TNBC [302]; however, this needs to be further investigated, as other studies did not find this correlation [58,269]. Increased protein expression of four genes, HER2_pY1248, PLK1, Rb_pS807/811, and EMA (MUC1), has been shown to be correlated with ONC201 resistance in TNBC, suggesting that inhibiting these genes may improve ONC201 efficacy [302]. The expression of CSC-related genes (e.g., ID1, CD44, HES7, and TCF3) correlates with ONC201 efficacy, and combining the expression of multiple genes is shown to be a potential predictive as well as PD biomarker [315]. In non-breast cancer settings, a high level of the c-myc protein is reported as a predictive biomarker for ClpP agonists in glioblastoma [316], and mutation of succinate dehydrogenase is proposed as a potential biomarker in neuroendocrine tumors [317].

Mitochondrial activity: Basal mitochondrial ATP% (e.g., the fraction OxPhos-derived ATP, as opposed to glycolysis-derived ATP) is proposed as a predictive biomarker of ONC212 in pancreatic cancer models [318], but not yet tested in breast cancers.

### 7.2. PD Biomarkers

Detecting mitochondrial dysfunction caused specifically by ClpP agonists in patients is needed to evaluate the drug efficacy. Detecting impaired OxPhos is feasible; however, specificity and sensitivity remain as major challenges. For instance, quantification of lactate/pyruvate can be used to detect OxPhos dysfunction [269,270]. However, absence of elevated lactate/pyruvate does not exclude the presence of OxPhos impairment, and the elevation of those metabolites may be condition dependent (e.g., exercise and acute crisis) [319]. Cytokines, such as fibroblast growth factor 21, growth differentiation factor 15 are stress response factors linked to mitochondrial function, and are proposed as diagnostic biomarkers of mitochondrial diseases; however, the specificity remains controversial [319,320,321].

Mitochondrial damage caused by ClpP agonists was successfully detected in vivo. Downregulation of TFAM, TUFM, PYCR2, HMGCS1, and mtDNA copy number was detected in breast tumors in mice treated with TR-107 [270]. However, mitochondrial dysfunction may trigger a compensatory protective response, making the assessment of specificity of ClpP agonists difficult [319]. Moreover, performing multiple tumor biopsies is not practical and often not available in clinic. Minimally or non-invasive monitoring of mitochondrial function in tumors is needed. The ClpP agonist increased NK cells in peripheral blood of prostate cancer patients, suggesting that NK cell activation might be useful as an indirect biomarker of drug activity [300,313].

### 7.3. Drug Combination

Combination therapy with ClpP agonists is likely to be needed to improve the therapeutic efficacy in breast cancer. Several combination strategies are currently being proposed and investigated as discussed below.

TRAIL: Most breast cancers are highly resistant to TRAIL, except for a subset of TNBC [322,323]. A synergistic effect of TRAIL and the ClpP agonist has been reported in breast and other cancer types. ONC201 downregulates multiple anti-apoptotic proteins, such as Mcl-1, Bcl-xL, XIAP [324], which modulate the TRAIL-mediated extrinsic apoptosis pathway [325,326]. Combining TRAIL with ONC201 converted the response from anti-proliferative to apoptosis in TRAIL-resistant non-TNBC cells and in vivo breast cancer models [301]. In addition, ClpP agonist-induced senescent TNBC cells are sensitized to TRAIL-mediated apoptosis [310].

Chemotherapy agents: In a study using a panel of human cancer cell lines including breast cancer cell lines, ONC201 was shown to be synergistic with multiple FDA-approved chemotherapeutic and targeted agents, such as azacitidine, bortezomib, dacarbazine, hydroxyurea, pralatrexate, sorafenib, topotecan, and vismodegib [296]. Taxanes (e.g., docetaxel and paclitaxel) have also been shown to synergize with ONC201 in a subset of TNBC cells [299].

PARP inhibitors: PARP inhibitors are relatively new class of agents showing an efficacy in BRCA-mutated breast cancer and TNBC [327]. The synergistic effect of PARP inhibitors (e.g., olaparib and rucaparib) with imipridone has been shown in vitro, but not yet in vivo [328].

Mitogen-activated protein kinase kinase (MEK) inhibitor: The combination of ONC201 and trametinib, a MEK inhibitor, showed a synergistic anti-tumor effect and increased in caspase 3/7 activity in TNBC [302].

The Enhancer of Zeste Homolog 2 inhibitors (EZH2i): EZH2 mediates gene silencing via H3K27 tri-methylation. Inhibition of EZH2 catalytic activity is shown to target a metastatic subpopulation in TNBC [329]. The combination of EZH2i and ONC201 showed a synergistic apoptosis in multiple breast cancer cell lines [303].

Histone deacetylase inhibitor (HDACi): HDACi are promising anti-cancer agents that inhibit cell proliferation of many types of cancer cells, and have been actively tested in clinical trials including breast cancers (see a review [330]). The combination of vorinostat (FDA-approved) and ONC201 had a synergistic effect on cell viability, and induced apoptosis in MCF7 cells [303]. Several HDACi have been exploited in combination with imipridones in CNS malignancies, and a synthetic lethality of combination of ClpP activation and HDACi is reported in glioblastoma [272,331].

Venetoclax/ABT-199: Synergistic anti-tumor effect of the combination of venetoclax and imipridones has been reported in AML models [312,332], but not yet in breast cancers.

Immunotherapy: Immunotherapies are revolutionizing the treatment of many cancers, including breast cancer. The clinical activity of programmed cell death-1/programmed death ligand-1 (PD-1/PD-L1) blockade by ICIs is shown with early- [12,333] and late-stage breast cancer patients [9]. The ICIs are currently being tested in combination with multiple therapies (e.g., chemotherapy, HER2-directed therapy, and CDK4/6 inhibitor) [334]. ONC201 has been shown to induce CD4/CD8+CD3+ T cell accumulation in colon cancer models, suggesting a that the combination of ClpP agonists with ICIs may be useful [300]. The combination of atezolizumab, an anti-PD-L1 antibody, and ONC201 is currently being tested in a clinical trial in endometrial cancer (NCT05542407). The combination of ClpP agonists with ICIs needs to be explored in breast cancers.

NK cells are anti-tumorigenic, innate cytotoxic lymphocytes, and are being investigated for use in breast cancers [335,336]. Peritumoral abundance of NK cells is shown to correlate with better pathological complete response (pCR) rate after neoadjuvant chemotherapy in large and locally advanced breast cancer [337]. ClpP agonists recruit and activate NK cells in breast and other cancer models [300,301], and the cytotoxicity of ClpP-agonist was enhanced by NK cells in breast cancer cell line [304].

While the combination of immunotherapy with a ClpP agonist is appealing, the impact of ClpP agonists on immune cells needs to be considered. For instance, T cell metabolism changes dynamically depending on the activation state [338]; Naïve T cells (quiescent) rely on OxPhos, and activated effector T cells primarily use glycolysis, while memory T cells and Treg are dependent on OxPhos and FAO [339,340]. Thus, breast cancer cells and T cells both rely on mitochondria. Therefore, defining similarities and differences between the cancer metabolism and immune cells is critical to determine potential metabolic crosstalk or competition within the TME. Such mechanistic insight may reveal novel strategies to inhibit tumor growth while maximizing anti-tumor immunity [341]. The function of NK cells also depends on mitochondrial dynamics and OxPhos [342], and is modulated by various metabolic conditions [343]. The presence of lactate in the TME is a major obstacle for cell-based immunotherapies as it impairs the cytotoxic abilities of T cells and NK cells. [344,345]. At present, our knowledge on the impact on ClpP agonists on immune system is limited. Understanding the impact of ClpP agonists on immune cells and TME is needed in future research.

## 8. Conclusions and Future Directions

Recent studies established that mitochondria are essential for breast cancer progression, and numerous efforts have been made to target mitochondria. ClpP agonists are an emerging class of mitochondria-targeting drugs featuring unique mechanism of action. Hyperactivation of ClpP leads to uncontrolled protein degradation in mitochondria. The impact of ClpP agonists goes beyond OxPhos inhibition; they broadly disrupt multiple mitochondrial functions (e.g., OxPhos, the TCA cycle, glutamine-proline metabolism, one-carbon metabolism, nucleotide biosynthesis, and the mevalonate pathway). ClpP agonist-induced mitochondrial dysfunction effectively halts cell growth and induces senescence, but not apoptosis in breast cancers. ClpP agonists show cytotoxicity in cancer models but not in non-malignant cells. More studies are required to understand the impact of ClpP agonists on immune cells and TME. The development of predictive biomarkers, PD biomarkers, and effective drug combination strategies is also needed. Regardless, further development of ClpP agonists is warranted.

## Figures and Tables

**Figure 1 cancers-15-01936-f001:**
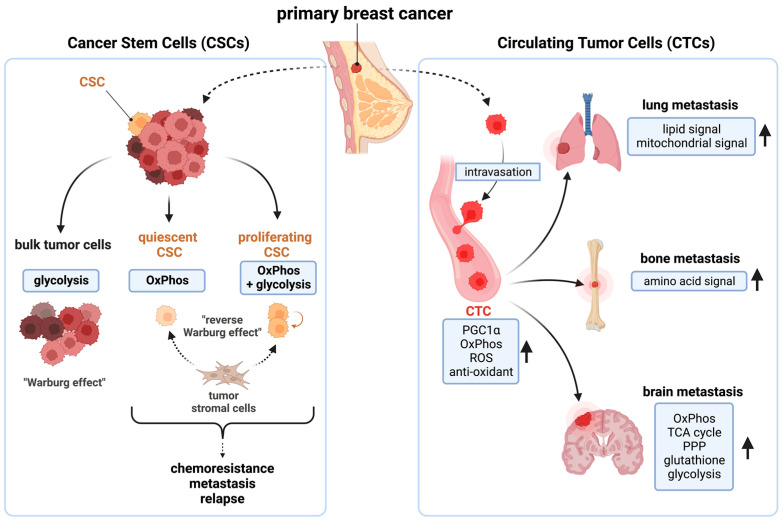
The function of breast cancer stem cells (CSCs) and circulating tumor cells (CTCs) are mitochondria dependent. (**Left**): while bulk cancer cells are glycolysis dependent, CSCs depend on OxPhos for survival. (**Right**): CTC-mediated distant metastases are also dependent on multiple mitochondrial functions. OxPhos: oxidative phosphorylation; PGC1α: peroxisome proliferator-activated receptor gamma coactivator 1-alpha, ROS: reactive oxygen species; TCA cycle: the tricarboxylic acid cycle; PPP: pentose phosphate pathway. This figure was generated by BioRender.com (19 January 2023).

**Figure 2 cancers-15-01936-f002:**
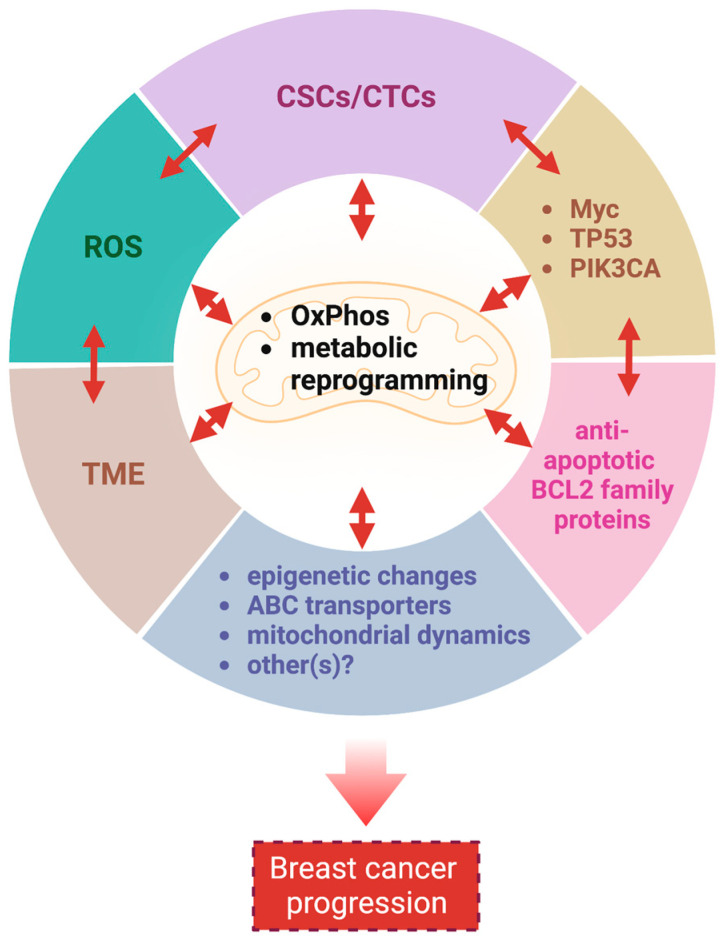
Mitochondria are critical for breast cancer progression. Mitochondria support breast cancer progression via multiple interrelated mechanisms. CSCs: cancer stem cells; CTCs: circulating tumor cells; ROS: reactive oxygen species; TME: tumor microenvironment; PIK3CA: phosphatidylinositol-4,5-bisphosphate 3-kinase catalytic subunit alpha; ABC: ATP-binding cassette. This figure was generated by BioRender.com (19 January 2023).

**Figure 3 cancers-15-01936-f003:**
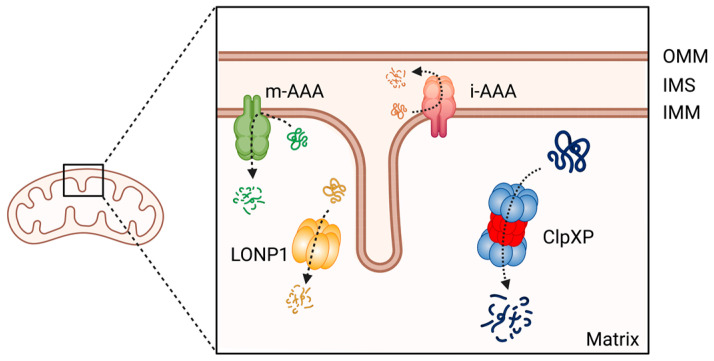
The member of AAA+ protease family in human mitochondria. ClpXP and LONP1 are located in the mitochondrial matrix, the i-AAA and m-AAA are located in the inner mitochondrial membrane. OMM: Outer mitochondrial membrane; IMS: intermembrane space; IMM: inner mitochondrial membrane. This figure was generated by BioRender.com (19 January 2023).

**Figure 4 cancers-15-01936-f004:**
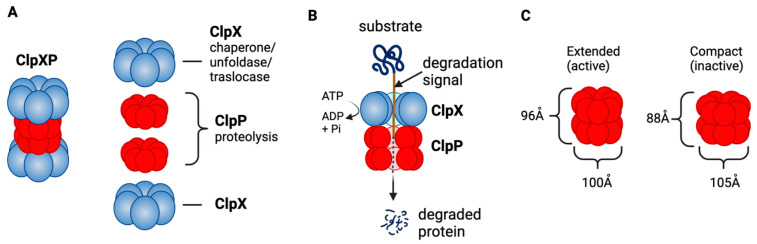
ClpXP structure and function as a mitochondrial protease. (**A**) ClpXP consists of ClpX (ATPase) and ClpP. (**B**) ClpX recognizes degrons of target protein, unfolds the proteins and threads them into ClpP for degradation. (**C**) Extended (active) and compact structure (inactive) of Staphylococcus aureus ClpP [243]. This figure was generated by BioRender.com (19 January 2023).

**Figure 5 cancers-15-01936-f005:**
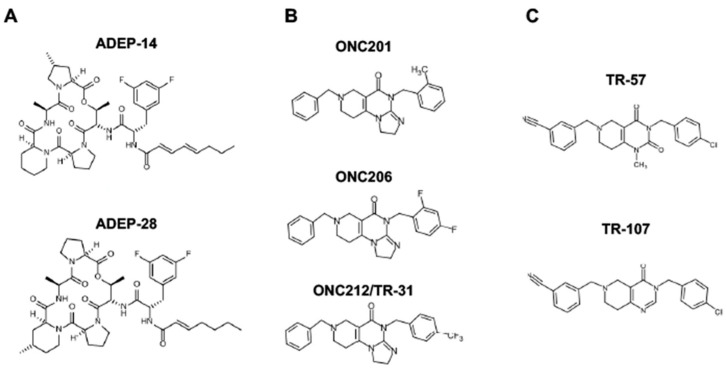
Representative ClpP agonists and structures. (**A**) Acyldepsipeptide antibiotics (ADEPs), (**B**) ONC201 and its analogs (imipridones), and (**C**) TR compounds.

**Figure 6 cancers-15-01936-f006:**
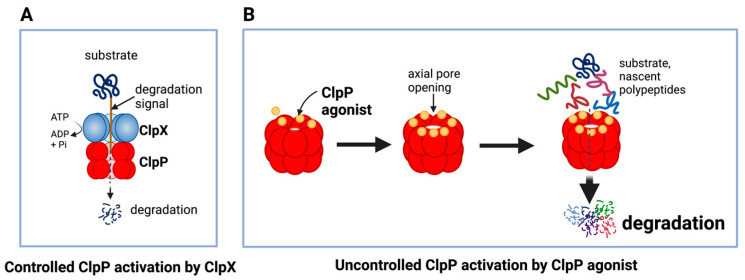
Controlled and uncontrolled proteolysis by ClpP. (**A**) In controlled proteolysis, target substrates are specifically recognized and unfolded by ClpX. The unfolded substrate is transferred into the barrel-like chamber of the ClpP, and proteolysis is carried out by active proteases at the inner surface of the chamber. (**B**) Uncontrolled proteolysis triggered by binding of ClpP agonists to ClpP. Binding of ClpP agonists (yellow spheres) to ClpP subunits displaces ClpX and triggers an opening of the entrance pore of the ClpP barrel, leading to unregulated degradation of nascent polypeptides and unfolded proteins. This uncontrolled ClpP activation does not require ClpX. This figure was generated by BioRender.com (20 January 2023).

**Figure 7 cancers-15-01936-f007:**
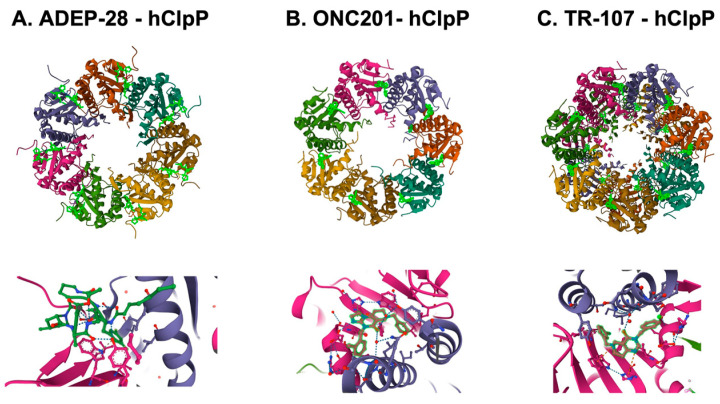
Crystal structures of human mitochondrial ClpP complex with ClpP ligands. (**A**) ADEP-28-hClpP binding [283] (PDB DOI: 10.2210/pdb6BBA/pdb), (**B**) ONC201-hClpP [271] (PDB DOI: 10.2210/pdb6DL7/pdb), (**C**) TR-107-hClpP [294]. (PDB DOI: 10.2210/pdb7UVU/pdb). Top panels show that each ClpP ligand (bright green) bound to ClpP. One heptamer ring for ADEP-28 and ONC201, or two heptamer rings for TR-107 are shown. Each subunit of ClpP is shown in a different color. Bottom panels show a magnified view of ligands (bright green) bound to the pocket between two ClpP subunits.

**Table 1 cancers-15-01936-t001:** Preclinical studies using ClpP agonists in breast cancers.

Compound	Breast Cancer Models (Cell Lines, In Vivo Models, Drug Administration)	Biological Effects	Drug-Resistant Models	Drug Combination/Drug Sensitivity Marker	Additional Note	Ref
**ONC201**	Cell lines: Breast and others in vivo: xenograft, female athymic nu/nu mice i.p. (50–100 mg/kg)	Induced production of TRAIL in tumor cells and TRAIL-mediated apoptosis, reduced growth and viability, anti-tumor effect in vivo			ONC201 penetration of the blood–brain barrier was suggested by TRAIL induction in brain tissue	[284]
**ONC201**	Cell lines: Breast and others no in vivo data with breast Oral (25 mg/kg), weekly (for non-breast models)	ONC201 synergizes with multiple anti-cancer drugs		sorafenib, azacytidine, bortezomib, dacarbazine, hydroxyurea, pralatrexate, sorafenib, topotecan, vismodegib		[296]
**ONC201**	Cell lines: Breast and others no in vivo data with breast i.v., 25 or 50 mg/kg (for non-breast models)	UPR, ISR, apoptosis, anti-proliferation	ONC201-resistant RKO (generated by exposure to ONC201)			[297]
**ONC201**	Cell lines: Breast no in vivo data	Inhibited cell growth, ISR, Mesenchymal-like and epithelial-like TNBC cells are sensitive. ONC201-resistant TNBC cells are defective of ISR induction	MDA231R-ONC201, MDA468R-ONC201 (generated by exposure to ONC201)			[298]
**ONC201**	Cell lines: Breast in vivo: MDA-MB-468 cells MFP xenograft in female athymic nude mice Oral, 50 mg/kg, once or 3 times per week	ISR, cell cycle arrest, anti-proliferation, but not apoptosis in most TNBC lines (6/8) tested. Activates caspase-8 cleavage, TRAIL-independent cell death in non-TNBC cells, synergized with taxanes in TNBC, anti-tumor effect in vivo	MDA-MB-436 and HCC1937 were relatively resistant in apoptosis	docetaxel, paclitaxel	pRb is important for ONC201 to arrest and maintain cells in the G1 phase. A decrease in cyclin D1 is a useful biomarker?	[299]
**ONC201**	Cell lines: Breast and others in vivo: MDA-MB231-luciferase xenograft in female athymic nu/nu mice 25, 50, or 100 mg/kg weekly, every 2, 3, or 4 weeks (oral or i.p.)	Inhibition of cell migration, invasion, metastasis via partially TRAIL-dependent mechanism, ISR, induced NK cell recruitment, activation, degranulation, induced CD4+/CD8+CD3+ T cell accumulation in syngeneic colon cancer model			NK cell numbers, activation, infiltration in post-Tx biopsies may yield useful correlative clinical information for drug efficacy	[300]
**ONC201**	Cell lines: Breast and others no in vivo data	Reduced growth and viability, ISR, OxPhos inhibition, mitochondrial damage, mtDNA depletion, no apoptosis	MDA-MB-231 rho0, UOK121 rho0, UOK262 FH mutant, Human Foreskin Fibroblast			[269]
**ONC201**	Cell lines: Breast in vivo: MDA-MB-436, 231, 361 MFP xenograft, female athymic nude mice, syngeneic E0771 model Oral (100 mg/kg), weekly	The combination ONC201 and rhTRAIL showed an anti-tumor effect, ONC201 triggered NK cells recruitment to tumors, NK cells kill breast cancer cells		rhTRAIL		[301]
**ONC201**	Cell lines: Breast (TNBC) no in vivo data	Growth inhibition, synergized with MEK inhibitor	HCC70, MDA-MB-157, SUM159 were resistant compared with CAL51, MDA-MB-468	MEK inhibitor trametinib/ ClpP level (higher ClpP protein expression correlated with higher sensitivity)		[302]
**ONC201**	Cell lines: Breast and others no in vivo data	apoptosis, ISR		EZH2 inhibitor (EPZ-6438 or PF-06821497), HDAC inhibitor (vorinostat)		[303]
**ONC201**	Cell line: BT474 no in vivo data	Inhibition of growth and intrinsic apoptosis and OxPhos, cell cycle arrest, ISR, depletion of mt-nucleoids and mtDNA, mitochondrial fragmentation, senescence-like phenotype, enhanced cell killing with NK cells		synergistic anti-cancer effect with NK cells or TRAIL		[304]
**ONC212/TR-31**	Cell lines: Breast and others no in vivo data of breast	Reduced cell viability in all cancer cells lines tested, anti-tumor effect in melanoma, liver cancers	non-malignant cells (WI38, HFF-1, MRC5, 18Co, 19Lu) were resistant		ONC201 and ONC206 were also used	[305]
**ONC212/TR-31**	Cell lines: Breast and others no in vivo data with breast	OxPhos inhibition, mitochondrial damage, reduced growth and viability, ROS induction	HEK293 T-REx CLPP KO	loss of ClpP showed ONC201 resistance	ONC201 and ADEP-1 were also used	[271]
**TR compounds**	Cell lines: Breast no in vivo data	ISR, reduced growth and viability, decreased mitochondrial protein, no apoptosis	siRNA-knockdown of CLPP in SUM159		ONC201 was also used	[268]
**TR compounds**	Cell lines: Breast in vivo: MDA-MB-231 cells MFP xenograft in female athymic nu/nu mice Oral, TR-107 (4 or 8 mg/kg, twice a day, twice a week)	Inhibited cell growth and viability, anti-tumor effect	MB231 CLPP KO, SUM159 CLPP KO		ONC201 and ONC206 were also used	[270]
**TR compounds**	Cell lines: Breast in vivo: MDA-MB-231 cells MFP xenograft in female athymic nu/nu mice Pre-treat cells with TR-57 (50 nM) or ONC201 (5 uM) for 48 h, then implanted to MFP	Inhibited cell growth and viability, inhibited multiple mitochondrial metabolism pathways, inhibited tumor initiation in vivo	MB231 CLPP KO, SUM159 CLPP KO, MCF-7 CLPP KO		ONC201 was also used	[58]
**TR compounds**	Cell lines: Breast no in vivo data	Inhibited cell growth	MB231 CLPP KO, HEK293 T-REx CLPP KO		ONC201 was also used	[294]

**Table 2 cancers-15-01936-t002:** Comparison of IC_50_ (nM) of ClpP agonists in representative breast cancer cell lines.

		Cell Lines (Subtypes)			
		MCF-7 (ER+)	MDA-MB-453 (HER2 Amplified)	MDA-MB-231 (TNBC)	SUM159 (TNBC)
ClpP agonists	ONC201	2400–25,000	768–3580	1000–10,000	32–20,000
	ONC206	1000–10,000		50–1000	180
	ONC212/TR-31	50–1000		<80	<80
	TR-57	25	10	17	14
	TR-107			23	12

These IC_50_ were summary of multiple preclinical studies [58,268,269,270,271,297,298,299,302,303,305].

**Table 3 cancers-15-01936-t003:** Pharmacokinetic analysis of select ClpP agonists.

Compound	Admin (mg/kg)	*T*_1/2_ (h)	*C*max	*C*max Unit	AUC	AUC Unit	F%	Animal Model	Ref.
ONC201	25, oral	6.42	44	uM	63.9	(0–∞; uM × h)	N.A.	C57/B6 mice	[284]
ONC212	125, oral	4.31	1.46	ug/mL	8.01	(0-t; ug/mL × h)	N.A.	C57/B6 mice	[305]
ONC201	2.0, i.v.	0.26	122 [0.31]	mg/mL [uM]	50.7	(0–∞; h × ng/mL)	N.A.	ICR mice	[270]
	10, oral	0.31	8.99 [0.023]		3.13 (1×)		1.2		
	25, oral	-	195 [0.50]		145 (46×)		N.A.		
ONC212/TR-31	2.0, i.v.	1.68	950 [2.2]		638		N.A.		
	10, oral	1.49	282 [0.64]		449		14		
TR-57	2.0, i.v.	1.52	1240 [3.0]		886		N.A.		
	10, oral	1.4	1700 [4.1]		2710 (866×)		61		
TR-107	10, oral	0.9	1440 [3.7]		2360 (754×)		N.A.		

*T*_1/2_: half-life; *C*max: maximal plasma concentration; AUC: area under the curve; F%: oral bioavailability. The numbers in the ( ) in the AUC are fold comparison relative to that of ONC201, 10 mg/kg (oral). N.A.: not available.

**Table 4 cancers-15-01936-t004:** Status of clinical trials using ClpP agonists in breast cancer patients.

NCT Number	Conditions	Intervention		Phase	Status	Results
NCT03394027	TNBC, Hormone Receptor Positive HER2 Negative Breast Cancer, Endometrial Cancer	Drug	ONC201	Ph2	Completes	not reported yet
NCT03733119	TNBC	Drug	ONC201	Ph2	Terminated early due to low number of enrollments	data for RR, PFS, OS, at 2 years not collected
		Dietary supplement	Methionine-Restricted Diet			

RR (relative risk), PFS (progression-free survival), and OS (overall survival).

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
