# Peer review of "Targeting Mitochondria with ClpP Agonists as a Novel Therapeutic Opportunity in Breast Cancer"

_cancers, 2023, doi:10.3390/cancers15071936_

Round 1

Reviewer 1 Report

Wedam and colleagues provide a sweeping review of mitochondria in breast cancer. The manuscript is extremely thorough and well organized. 

I have found that the manuscript is not reviewable at this point- it includes 408 references and is 30 pages long, not including references. The title and abstract imply it focuses on ClpP agonists, but this only comprises a small proportion of the manuscript. ClpP is not discussed until section 5- line 579. In addition there is a sweeping review of mitochondria and singaling in breast cancer and all types of mito-targeting drugs in  breast cancer.

This topic is of great interest but currently is manuscript is simply too long to be useful, or readable, by any typical audience.

Reviewer 2 Report

The manuscript entitled “Targeting mitochondria with ClpP agonist as novel therapeutic 2 opportunity in breast cancer" by Rohan Wedam and Colleagues, reports on an update of mitochondrial roles in breast cancers to elucidate how mitochondria can represent a valuable therapeutic target in breast cancer. The Authors outline the use of mitochondria-targeting drugs in breast cancers and highlight the ClpP agonists, as emerging mitochondria-targeting drugs in a combination with other drugs.

Overall, The Authors have prepared an impressively comprehensive review to elucidate why functional mitochondria are essential for breast cancer cells and how targeting mitochondria with ClpP agonist represents the better approach in breast cancer. In addition, the authors have provided different examples of metabolic, genetic and epigenetic mitochondria-dependent changes seen in breast cancers. The tables detailing the status of mitochondria-targeting therapies (with a focus on ClpP activating drugs) in breast cancers and relevant references will be extremely useful resources for researchers and for clinicians interested in this field.

Please, see below for comments about scientific content, including figures and minor editorial comments to improve readability. I suggest minor revisions as below.

I listed down some points to be clarified or improved:

Simple summary: 

1. Line 18: I suggest specifying that uncontrolled protein degradation refers to the degradation of mitochondrial matrix proteins or in general ClpP substrates.

2. Introduction: line 80: Please define “the mitochondrial caseinolytic protease ClpP” as “Caseinolytic Mitochondrial Matrix Peptidase Proteolytic Subunit (ClpP)” at first use

3. Line 114: The author says “A breadth of evidence shows that mitochondria are not defective, rather essential to cancer cells [44]”. Please consider adding other references to sustain this breadth of evidence. Consider citing e.g. Liu et Shi, Mitochondria as a target in cancer treatment, 2020.

4. Please, insert space between the words and the references all over the text

5. define also CSCs and CTCs in the legend of Fig. 1

6. reference 64 is not related to the topic

7. Line 152: Please add “-“ for “CSC-like cells”. Is ref. 64 correct? Please check. 

8. Line 199: ‘Glutaminase (GLS), a mitochondrial enzyme converts glutamine to glutamate’….something is missing

9. Line 321: change into GLUD/GDH1

10. Line 338: text size is increased

11. Figure 1: Please also define CSCs and CTCs in the figure legend. 

12. Figure 2: Line 321: Please replace GLUD with GLUD/GDH1 as shown in the figure.

13. Figure 2: Line 338: Please check the text size.

14. Line 377: Please define “PI3K” as phosphatidylinositol 3-kinase” at first use.

15. Line 381: Please define “mTOR” as a “mechanistic target of rapamycin” at first use.

16. Line 402: Please, consider re-evaluating the title considering the dual role of ROS. The Author says “ Breast CSCs have a lower level of ROS compared with bulk tumours, and pharmacologic inhibition of ROS scavengers in breast CSCs markedly decreased their clonogenicity[212].” In this context, ROS aren’t pro-tumorigenic. 

17. Figure 3: Please, explain the abbreviations in the figure legend.

18. Line 415: Please, explain the acronym MAPK at first use. MAPK stands for Mitogen-activated protein kinase

19. Line 515: the concentrations at which metformin inhibits Cpxl are very high; perhaps, this molecule’s activity does not have much to do with the complex. Please, I ask the authors to extend this part.

20. Line 525: Please use the bold for “Devimistat”.

21. Line 553: Please delete the comma after “NCT05358639”

22. Line 536: Please add reference 287 for TVB-2640.

23. Line 581: Please, define better and more precisely the mitochondrial proteome

24. Line 587: Please merge the references in the square brackets. 

25. Figure 4, Line 602: Please check the text size.

26. Line 605: The acronym ClpP should be explained only at first use in line 80. 

27. Line 634 and 674: Please delete the round bracket. 

28. Line 707: ONC201 doesn't eliminate completely but the reference says it's important. It doesn't eliminate completely, but the reference says it's important. And anyway in glioblastoma, this effect seems to be very important. The authors, even if they focus on one type of cancer, should highlight these effects. doi:10.1016/j.neo.2017.10.002.

29. Line 716, 790 and 863 and 971: Please replace “Tfam” with “TFAM”. 

30. Line 1014: Please correct the meaning of the MEK acronym and replace it with “Mitogen-activated protein kinase kinase”

31. Please check all the text and insert the space before the references. 
